# Flux of Root-Derived Carbon into the Nematode Micro-Food Web: A Comparison of Grassland and Agroforest

**Christin Hemmerling [1],\*, Zhipeng Li [2], Lingling Shi [3,4], Johanna Pausch [5] and Liliane Ruess [1]**

1   Institute of Biology, Ecology, Humboldt Universität zu Berlin, Philippstraße 13, 10115 Berlin, Germany; liliane.ruess@biologie.hu-berlin.de
2   J. F. Blumenbach Institute of Zoology and Anthropology, University of Göttingen, Untere Karspüle 2, 37073 Göttingen, Germany; lizhipengucas@gmail.com
3   Key Laboratory for Plant Diversity and Biogeography of East Asia, Chinese Academy of Sciences, Kunming Institute of Botany, Heilongtan, Kunming 650201, China; shilingling@mail.kib.ac.cn
4   Department of Agricultural Soil Science, University of Göttingen, 37707 Göttingen, Germany
5   Department of Agroecology, Bayreuth Center of Ecology and Environmental Research (BayCEER), University of Bayreuth, Universitätsstraße 30, 95440 Bayreuth, Germany; Johanna.Pausch@uni-bayreuth.de
*   Correspondence: christin.hemmerling.1@hu-berlin.de

**Abstract:** Carbon (C) cycling is crucial to agroecosystem functioning. Important determinants for the belowground C flow are soil food webs, with microorganisms and microfaunal grazers, i.e., nematodes, as key biota. The present study investigates the incorporation of plant-derived C into the nematode micro-food web under two different cropping systems, grassland (ryegrass (*Lolium perenne* L.) and white clover (*Trifolium repens* L.)) and agroforest (willow (*Salix schwerinii* Wolf and *Salix viminalis* L)). To quantify the C flux from the plant into the soil micro-food web, grass and willow were pulse-labeled with $^{13}CO_2$ and the incorporation of $^{13}C$ into the nematode trophic groups was monitored 3, 7, 14 and 28 days after labeling. The natural stable isotope signals ($^{13}C/^{12}C$, $^{15}N/^{14}N$) were analyzed to determine the structure of the nematode micro-food web. The natural isotopic $\delta^{15}N$ signal revealed different trophic levels for omnivores and predators in grassland and agroforest soils. The incorporation of plant C into nematode tissue was detectable three days after $^{13}CO_2$ labeling with the highest and fastest C allocation in plant feeders in grassland, and in fungal feeders in agroforest soil. C flux dynamics between the aboveground vegetation and belowground micro-food web varied with cropping system. This demonstrates that crop-specific translocation of C affects the multitrophic interactions in the root environment, which in turn can alter soil nutrient cycling.

**Keywords:** C flux; root-derived C; plant type; agroforest; nematodes; micro-food web

## 1. Introduction

Soils are major carbon (C) pools and store about two-thirds of the global soil C in an organic form [1]. About 20% of the C fixed by crops is transferred to belowground pools [2]. This plant-derived C reaches the soil either via the decomposition of plant litter, by lysates of damaged cells or root exudates secreted by living root cells [3]. Especially, rhizodeposits consisting of easily degradable organic compounds are the primary source of energy for the soil food web [4]. The transfer of C from the crop into the soil is affected by plant type [2]. Changes in land use lead to changes in the soils' capacity to store C [5–7], with the greatest impact attributed to conversion of croplands to native vegetation and vice versa [8].

Perennial crops are characterized by a year-long vegetation cover and greater root biomass, thus contributing more organic C to the soil, providing higher energy inputs for biological activity than annual agricultural systems [9]. Grasslands translocate up to 30% of plant C belowground, resulting in a large pool of labile organic C available for soil biota, while the major C sink in trees is not the root but rather the stem [10]. However, depending

on tree age and growth phase, leaves and fine roots can act as alternative C sinks, thereby changing the C flux within the tree [11,12]. Further, associations with mycorrhizal fungi influence the translocation of recently fixed plant C to the soil microbial community and subsequently the uptake of root-derived C by microbial feeders [13].

The deliberate combination of annual crops with perennial grasslands or trees on the same arable land is an example of agroforestry and the most common method in Germany [14,15]. Typically, strips of fast-growing tree species such as poplars (*Populus* spp. L.) and willows that yield a high amount of woody biomass in a short time are integrated into conventional agricultural sites [16]. After a rotation period of two to ten years, the trees are harvested for the production of bioenergy [17]. Willows are the most suitable energy crops as they have a higher net energy output and the lowest greenhouse gas emissions compared to annual crops used for biofuels, e.g., wheat, rapeseed, and sugar beet [18]. Grasses intercropped with legumes, e.g., clover provide a suitable biogas substrate are also used due to high biomass production and low lignin content [19]. The combination of willows and grass-clover mixtures is thus energetically highly efficient. On the other hand, the environmental benefits of short-rotation agroforestry include reductions in soil erosion and leaching of soil nutrients, improvement of C-sequestration and enhancement of biodiversity [20].

The type of aboveground vegetation shapes the trophic structure of belowground nematode communities: a meta-analysis revealed that fungivore nematode abundance and biomass increases from grassland to cropland and forest, while the opposite pattern can be observed for herbivore nematodes [21]. On the other hand, bacterivore nematodes are the predominant trophic group on a global scale [22]. Omnivores and predators with long life cycles and sensitivity to disturbances are more abundant in mature areas than newly established habitats in grassland [23], as well as forest ecosystems [24].

In soil food webs, C is channeled either along the herbivore or detritivore food chain: While the herbivore food chain is based on primary production and fuels the root C and energy channel, the detritivore food chain is related to the decomposition of dead organic matter and comprises the fungal and bacterial energy channel [25]. In systems with a low substrate C/N ratio (i.e., high quality, easier to decompose), e.g., in grasslands, C is mainly transferred along the bacterial decomposition pathway, while in systems with a high substrate C/N ratio, e.g., forests, decomposition processes are dominated by fungi [26]. Furthermore, due to differing decomposition rates, the bacterial C and energy channel is referred to as fast cycle and the fungal channel as slow cycle [27,28].

One animal group that occupies key positions within the soil food web are nematodes [21]. Both plant parasites as well as beneficial free-living nematodes interact in various ways with crop plants, thereby affecting soil C and nutrient dynamics. Plant-parasitic nematodes impose a C loss on plants either directly via feeding on roots or indirectly via the leakage of cell metabolites from damaged root cells [29]. Further, nematodes regulate microbial composition and decomposition activity by grazing on bacteria and fungi [30]. The subsequent excretion of $NH_4^+$ and amino acids makes these nutrients, which are otherwise bound in microbial biomass, accessible for plant uptake [31].

As their activity is regulated by biological and physical soil conditions [29], using nematodes as bioindicators provides a tool for the assessment of soil quality and functioning, especially with regard to land use changes [32]. More precisely, analyzing the presence and abundance of nematode functional guilds (i.e., similar feeding types and life strategies) makes it possible to assess food web structure and condition, i.e., basal, structured or enriched [33]. Applying this faunal analysis concept, it has been shown that the functional succession from bacterivore to fungivore nematodes in an arable soil was driven by the input of high C/N resources [27]. In line with this, the fungivore to bacterivore ratio (*f/b* ratio) increased following reforestation on degraded soil covering Karst terrain [34].

The present study investigates the incorporation of plant-derived C into the nematode micro-food web under two different perennial land use types, i.e., grassland and agroforest. The objectives were to determine the predominating soil C and energy channels under both land use types, to uncover plant type-dependent differences in the trophic

structure of the nematode micro-food webs and to examine the overall plant C flux into the nematode community.

## 2. Materials and Methods

### 2.1. Study Site

The study site was located in Reiffenhausen (51°39′83″ N, 9°98′75″ E; 325 m a.s.l.) south of Göttingen in Central Germany (Figure S1). The region is characterized by a temperate climate, with both western maritime Atlantic and eastern continental influences. Mean average temperature is 9.2 °C and mean annual precipitation is 642 mm, with 312 mm during the vegetation period from May to September [35].

The study area is characterized by sedimentary deposits of Middle and Upper Triassic sandstone material, partly mixed with clay stone material and covered by loess sediments [36]. The soil types present are Stagnic Cambisol and Haplic Stagnosol, while the soil texture varies from loamy sand in the northeastern part to silty loam in the southwestern part of the site [36].

### 2.2. Experimental Design and Soil Properties

The experimental plots were established on former arable land in March 2011, with winter barley (*Hordeum vulgare* L.) as the preceding crop. The study side comprises the two cropping systems, grassland and agroforest, planted in a strip design. The grass strips (*n* = 3) are each 9 m wide and 75 m long, and consist of a grass–clover mixture (dominated by *Lolium perenne* L. mixed with *Trifolium repens* L.) [37]. The grassland represents a low-input system without fertilizer application. However, grass cutting was performed three times per year [38]. The willow strips (*n* = 3) are each 7.5 m wide and 75 m long and consist of *Salix schwerinii* Wolf and *Salix viminalis* L. During the strip installation the willow cuttings were planted in double rows with a distance of 1.5 m between rows and a distance of 0.75 m between cuttings within a row [35]. At the time of sampling, the willow trees were three years old and about 4 m high. The willow strips function as a short-rotation forestry plantation, i.e., a low-input system with a fast-growing energy plant. One rotation cycle lasts three years without fertilizer application. Combining the production of woody biomass as energy feedstock with conventional field crops is an increasingly popular management practice in Germany [15].

The proportions of clay, silt and sand under grassland were 10%, 45% and 45%, respectively. The soil pH was almost neutral at 6.6, bulk density was 1400 kg m$^{-3}$ [39] and microbial C ($C_{mic}$) 2700 g kg$^{-1}$. The soil under willows consists of 15% clay, 40% silt and 45% sand [37]. The pH was 5.6, while bulk density and $C_{mic}$ were 1400 kg m$^{-3}$ and 2400 g kg$^{-1}$, respectively [35]. Measurements were taken in 0–30 cm soil depth for pH and bulk density, and 0–15 cm for $C_{mic}$.

### 2.3. Pulse Labeling with $^{13}CO_2$

The pulse labeling took place on 20 June 2017 for willow and on 22 August 2017 for grass. At each plant type, five labeling chambers were installed (Figure S1). For the willow labeling, one strip was selected and chambers were installed at a distance of at least 10 m from each other. Each chamber contained one willow tree with the upper branches bent to fit the height of the chamber. In the grass strips, plots were established as blocks (9.0 × 6.5 m) differing in experimental treatment (e.g., with/without fertilization), not allowing installment of five labeling chambers within the same strip. Therefore, the chambers were set up in two blocks with the same treatment (no fertilizer application). Two chambers were installed in the grass strip next to the selected willow strip (block one) and three in a second grass strip located 50 m away (block two; Figure S1).

Each labeling chamber consisted of a stainless-steel frame (1 × 1 m), which was inserted into the soil to a depth of 10 cm, and an aluminum frame on top. The height of the aluminum frame was 1 m for grass and 2 m for willow. The frame was covered with translucent LDPE (low-density polyethylene) foil. The chambers contained cooling packs

to prevent high transpiration by the plant and fans to ensure permanent $CO_2$ dispersal. In the willow plot, understory plants were removed and the ground of each chamber was covered with black plastic foil to prevent $^{13}C$ assimilation and incorporation into the system by herbaceous plants.

For the $^{13}C$ labeling, a plastic beaker with 20 g $Ca^{13}CO_3$ as tracer was installed in each chamber. Every two hours, 50 mL of hydrochloric acid (HCl) was applied to the tracer to enable the production of $^{13}CO_2$. The released $^{13}CO_2$ circulated inside the chamber for 6 h, then the labeling process was ended and the foil and frame were removed.

### 2.4. Plant, Soil and Nematode Sampling

Samples of the soil, plants and nematodes were taken 3, 7, 14 and 28 days after the labeling of the grass or willow, respectively. Additionally, unlabeled samples were taken in neighboring grass and willow strips, in which no labeling chambers were installed, to determine the natural stable isotope ratios of $^{13}C/^{12}C$ and $^{15}N/^{14}N$ in plant ($n = 4$) and nematode tissue ($n = 5$) and in the soil ($n = 5$) under grass and willows. Per labeling chamber, one soil sample, one nematode sample and one sample of the plant (divided in root and shoot subsamples) were analyzed, resulting in 5 replicates per day and plant type for each sampled material.

Plant subsamples were randomly collected and represented about 20% of the total plant biomass. For willow, the shoot subsample comprised leaves and twigs and the root subsample a mixture of coarse and fine roots. The soil was sampled with a split tube at 0–15 cm depth and sieved to 2 mm. Soil and plant material were dried at 60 °C for 48 h, weighed, milled and stored in a desiccator until stable isotope analysis.

The nematode sampling was carried out with a stainless-steel corer (diameter 8 cm, depth 15 cm). Of these 30 g fresh weight (FW) of each sample were used to determine the soil dry weight (DW) and 50 g FW were subjected to heat-extraction of nematodes after a modified Baermann method [40]. The extraction procedure started at room temperature (20 °C) for 24 h, followed by a gradual heating in 5 °C steps for 6 h, beginning at 20 °C and ending at 45 °C. Afterwards, nematodes were fixed in 4% formaldehyde solution.

The total number of nematodes in a sample was counted, the trophic level determined using the identification key proposed by Bongers [41] and individuals assigned to plant, fungal, and bacterial feeders, omnivores and predators, according to Yeates et al. [42]. In samples which contained more than 1000 individuals, 10% of individuals were identified, whereas in samples with less than 1000 individuals, 100 individuals were inspected. Dauerlarvae of the family Rhabditidae were not included in the following analyses as they represent a non-active resting stage [32].

### 2.5. Stable Isotope Analysis

2.5.1. Plant and Soil

The plant and soil samples were placed into tin capsules for the measurement of relative N and C isotope abundances using an elemental analyzer (EA) NA1500 (Carlo Erba Instruments, Milano, Italy) coupled to a Delta plus isotope ratio mass spectrometer (IRMS) (Thermo Fisher Scientific, Bremen, Germany) through a ConFlo III interface (Thermo Electron Corporation, Bremen, Germany).

2.5.2. Nematodes

As preparation for stable isotope analysis, nematodes were sorted by trophic group using an inverse light microscope and stored in glass vials. Before further processing, nematode samples were subjected to a washing procedure with deionized water to remove the formaldehyde from the animal tissue. The supernatant of each sample was removed so that 500 μL of the sample remained within the vial. Then, 500 μL of deionized water were added. This procedure was repeated after one hour. Afterwards, the samples were stored overnight in a refrigerator to allow further exchange of water and formaldehyde in the nematode tissue. The following day, the entire supernatant of each sample was

removed, the nematode individuals were transferred into silver capsules (4 × 6 mm; IVA Analysetechnik GmbH, Meerbusch, Germany) and dried at 60 °C for six days. Samples for natural variations of isotopes and labeled samples were dried in different drying cabinets to prevent cross-contamination. The weights of empty silver capsules and of dried nematode tissue of each sample were measured using an ultra-micro balance (Mettler Toledo XP6U Comparator, Mettler Toledo, Colombus, OH, USA).

A high-sensitivity elemental analyzer/isotope ratio mass spectrometer setup (μEA/IRMS) was used to perform the measurements of stable isotope ratios. It allowed an accurate analysis of small sample amounts, i.e., up to 1 μg of C or 0.6 μg of N with < 1‰ standard error after blank correction [43]. The EA (Eurovector, Milano, Italy) was equipped with a Blisotec low blank autosampler (Blisotec, Jülich, Germany) and contained smaller oxidation and reduction reactor tubes (i.d. 7.8 mm, o.d. 10.5 mm, length 450 mm) to obtain sharper sample peaks in the mass spectrometer [43]. Further, the EA was coupled to a Thermo Delta Vplus isotope ratio mass spectrometer via a ConFlo IV interface (both from Thermo Fisher Scientific, Bremen, Germany). V-PDB (Vienna Pee Dee belemnite) and atmospheric $N_2$ were used as standards for $^{13}C$ and $^{15}N$, respectively, and acetanilide for internal calibration.

Isotope natural abundance was calculated using the δ notation [‰] relative to a standard (V-PDB and $N_2$ Air) calculated as $\delta_n E\ [‰] = [(R_{sample} - R_{standard})/R_{standard})] \times 1000$, where $E$ is the element, $n$ the weight of the heavier isotope and $R$ the ratio of heavy to light isotopes. The isotopic fractionation was calculated as $\Delta \delta_n E\ [‰] = \delta_n E_{comsumer} - \delta_n E_{food}$. Isotopic incorporation of $^{13}C$ was calculated as the shift in the isotopic signature between labeled and unlabeled control samples as $\Delta^{13}C\ [‰] = \delta^{13}C_{labeled\ sample} - \delta^{13}C_{unlabeled\ sample}$.

### 2.6. Statistical Analysis

The software R (version 4.0.4. "Lost Library Book") was used for statistical analyses.

The Kruskal–Wallis rank sum test was performed on isotope natural abundances of nematodes, roots shoots, and the soil utilizing Dunn's post hoc test with Bonferroni correction (significance level at $p < 0.05$).

Data concerning the incorporation of $^{13}C$ into nematode tissue expressed as $\Delta^{13}C$ were subjected to analysis of variances (one-way ANOVA) utilizing Tukey's HSD post hoc test (significance level at $p < 0.05$).

The incorporation of $^{13}C$ into nematode tissue was compared between sampling days and trophic groups using linear mixed effects models for each plant type. Sampling days and trophic groups were set as fixed effects. The investigated different spatial scales were nested in a hierarchical structure as follows: "soil core" nested within "chamber", "chamber" nested within "plant type". Further, "chamber" and "soil core" nested within "chamber" were set as random effects. $\Delta^{13}C$ values of basal resources and the soil were compared between sampling days and plant type with "chamber" and "soil core" nested within "chamber" as random effects. Since chambers in the grass plots were distributed in two blocks, "block" was set as random effect in the models of grass as well.

Differences in the incorporation of $^{13}C$ into trophic groups between plants were analyzed using the two-sample t-test using data of all sampling dates together (i.e., 3, 7, 14 and 28 days after $^{13}CO_2$ labeling).

### 3. Results

#### 3.1. Nematode Density and Trophic Structure

The nematode population density across all samplings, i.e., over a period of 28 days, was similar under both land use types (Table 1). Plant feeders were the predominant trophic group in grassland (47.7 ± 11.8%) as well as in agroforest soils (40.0 ± 16.0%). Plant type-specific differences were most pronounced for microbial grazers: in grassland, bacterial feeders occurred more than twice as often as fungal feeders, with 30.6 ± 9.7% and 12.4 ± 4.7%, respectively. On the other hand, in agroforest soils the proportions of microbial feeders were equal, with 26.5 ± 12.6% for both fungal and bacterial feeders.

Accordingly, the fungal feeder/bacterial feeder ratio (*f/b* ratio) was three times lower under grass compared to willow. However, due to the strong variation in nematode distribution, the reported changes in nematode community structure are not significantly different. Omnivores and predators were scarce and not affected by land use type.

**Table 1.** Mean nematode population density [Ind. $g^{-1}$ DW $\pm$ s.d.], proportion of trophic groups [% $\pm$ s.d.] as well as the fungal to bacterial feeder ratio (*f/b* $\pm$ s.d.) across all sampling days in grassland and agroforest soils.

|  | **Grassland** | **Agroforest** |
| --- | :---: | :---: |
| Population density (Ind. $g^{-1}$) | 21.3 $\pm$ 8.0 | 19.5 $\pm$ 8.8 |
| Trophic groups (%) |  |  |
| Plant feeders | 47.7 $\pm$ 11.8 | 40.0 $\pm$ 16.0 |
| Fungal feeders | 12.4 $\pm$ 4.7 | 26.1 $\pm$ 14.7 |
| Bacterial feeders | 30.6 $\pm$ 9.7 | 26.8 $\pm$ 10.3 |
| Omnivores | 6.5 $\pm$ 4.4 | 5.4 $\pm$ 3.9 |
| Predators | 2.8 $\pm$ 2.6 | 1.6 $\pm$ 1.5 |
| *f/b* ratio | 0.4 $\pm$ 0.2 | 1.3 $\pm$ 1.0 |

### 3.2. The Nematode Micro-Food Web

3.2.1. Natural Stable Isotope Ratios of Soil and Basal Food Web Resources

The $\delta^{13}$C signatures of the soil in grassland ($-27.3 \pm 0.5$‰) and agroforest ($-27.8 \pm 0.2$‰) were higher than the $\delta^{13}$C values of the corresponding shoots (grass: $-31.5 \pm 1.0$‰, willow: $-28.5 \pm 0.2$‰) and roots (grass: $-30.5 \pm 0.6$‰, willow: $-28.0 \pm 1.1$‰), respectively (Figure 1). This was significant for grassland soil compared to shoots ($p < 0.01$). Similarly, the $\delta^{15}$N signatures of the soil with $5.6 \pm 0.7$ and $4.7 \pm 0.2$‰ for grass and willow, respectively, were higher than the $\delta^{15}$N signatures of the shoots (grass: $4.4 \pm 2.5$‰, willow: $-0.9 \pm 0.8$‰) and the roots (grass: $1.7 \pm 0.9$‰, willow: $1.0 \pm 0.5$‰) (Figure 1). This was significant for soil compared to roots in grassland ($p < 0.05$) and to shoots in agroforest ($p < 0.01$).

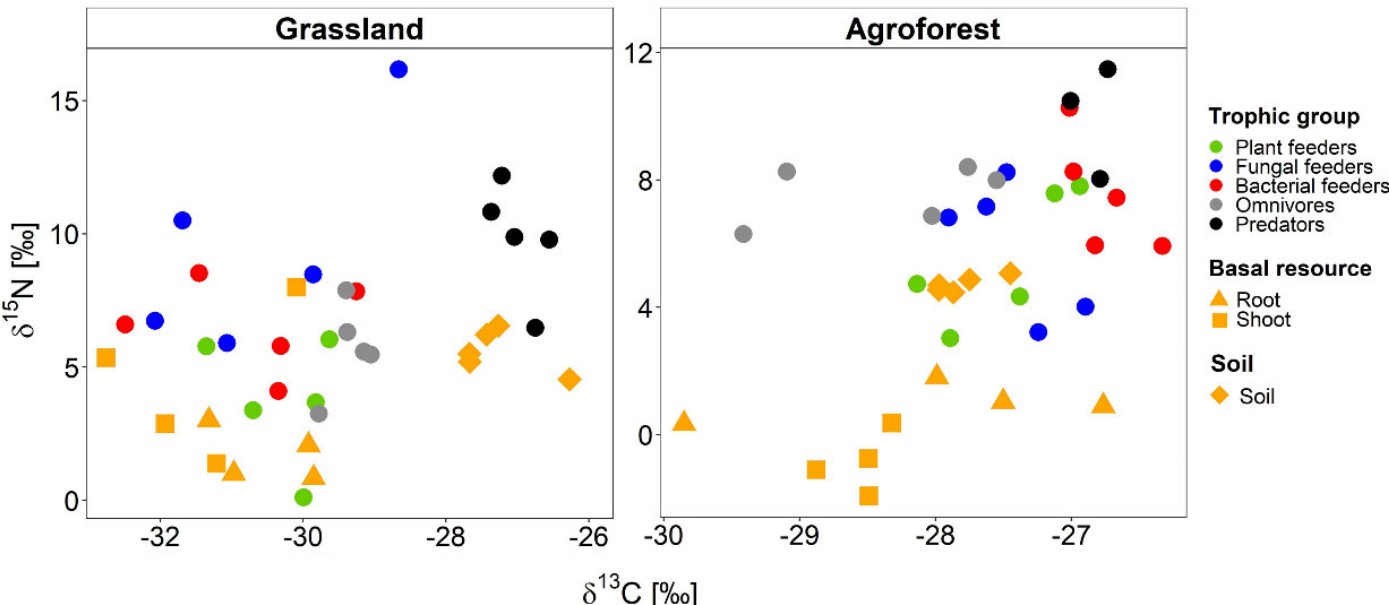

**Figure 1.** Natural stable isotope signatures of C (‰ $\delta^{13}$C) and N (‰ $\delta^{15}$N) of the soil, the basal plant resources shoots and roots, and in nematode trophic groups in grassland and agroforest systems.

### 3.2.2. Natural Stable Isotope Ratios of Nematode Trophic Groups

The $\delta^{13}$C signatures of nematodes ranged from $-32.5$ to $-26.6$‰ in grassland and from $-29.4$ to $-26.3$‰ in agroforest (Figure 1). In grassland soil, predators showed the highest $\delta^{13}$C values ranging from $-27.4$ to $-26.6$‰, which was significantly higher ($p < 0.05$) compared to fungal, bacterial and plant feeders with $\delta^{13}$C signatures ranging from $-32.5$ to $-29.3$‰, $-32.1$ to $-28.7$‰ and $-31.4$ to $-29.6$‰, respectively. Omnivores had the narrowest range in $\delta^{13}$C with $-29.8$ to $-29.1$‰.

In contrast, in agroforest soil, the omnivores showed the widest range of $\delta^{13}$C with $-29.4$ to $-27.6$‰ compared to all other trophic groups (Figure 1). Bacterial feeders and predators had the lowest $\delta^{13}$C values with $-27.0$ to $-26.3$‰ and $-27.0$ to $-26.7$‰, respectively. Moreover, the $\delta^{13}$C signatures of omnivores under willow were significantly lower compared to those of bacterial feeders ($p < 0.01$) and predators ($p < 0.05$).

The $\delta^{15}$N values in nematode tissues spanned over a much wider range in grassland with 0.1 to 16.2‰ compared to agroforest with 3.0 to 11.5‰. In grassland soil, fungal feeders and predators had high $\delta^{15}$N values ranging from 5.9 to 16.2‰ and from 6.5 to 12.2‰, respectively. Plant feeders displayed low $\delta^{15}$N values with 0.1 to 6.0‰, which were significantly lower ($p < 0.05$) compared to predators. In agroforest soil, predators showed high $\delta^{15}$N signatures ranging from 8.0 to 11.5‰. Plant feeders and fungal feeders had a low $\delta^{15}$N content with 3.0 to 7.8‰ and 3.2 to 8.2‰, respectively, while bacterial feeders hold an intermediate position with 5.9 to 10.3‰.

### 3.2.3. Isotopic Fractionation across Trophic Levels

Considering the shifts in $^{13}$C of nematode trophic groups compared to basal food web resources, in grassland, omnivores were associated to roots as assigned by an enrichment of $\Delta^{13}$C of $1.2 \pm 0.3$‰ (Table 2). Compared to roots, microbial grazers showed a weak depletion in $^{13}$C between $-0.3 \pm 1.2$‰ and $-0.1 \pm 1.4$‰ for bacterial and fungal feeders, respectively. Predators were clearly separated from the isotopic signal of roots by an enrichment of $^{13}$C of $3.5 \pm 0.3$‰. Further, a distinct fractionation of more than 3‰ occurred between predators and plant feeders ($3.3 \pm 0.9$‰), fungal feeders ($3.7 \pm 1.6$‰) and bacterial feeders ($3.8 \pm 0.9$‰). In contrast, the isotopic signal of omnivores was closer to that of plant, fungal and bacterial feeders with a $\Delta^{13}$C of $0.9 \pm 0.5$‰, $1.3 \pm 1.3$‰ and $1.4 \pm 1.3$‰, respectively.

In the agroforest, the $\delta^{13}$C values of all nematode trophic groups (except omnivores) were linked to the isotopic signal of roots (Table 2). Plant and fungal feeders displayed the closest association with roots by a $\Delta^{13}$C of $0.5 \pm 0.5$‰ and $0.6 \pm 0.4$‰, respectively, while bacterial feeders showed a higher fractionation with a $\Delta^{13}$C of $1.3 \pm 0.3$‰. The isotopic signal of omnivores was slightly depleted in $\Delta^{13}$C of $-0.3 \pm 0.8$‰ compared to root tissue. Moreover, omnivores were clearly separated from microbial grazers as well as plant feeders with a $\Delta^{13}$C of $-1.6 \pm 0.6$‰ to $-0.9 \pm 1.0$‰. On the other hand, predators were related to fungal feeders and plant feeders with a $\Delta^{13}$C of 0.5‰.

The $^{15}$N enrichment in nematodes compared to the dominant plant resource (i.e., roots) revealed that in grassland soil, omnivores showed the closest relation to roots with a $\Delta^{15}$N of $4.0 \pm 1.7$‰ (Table 2). All other nematode trophic groups separated from the isotopic signal of the roots. In agroforest soil, all nematode feeding types showed a distinctly higher $^{15}$N signal compared to roots, with a $\Delta^{15}$N ranging from $4.5 \pm 2.1$‰ in plant feeders to $9.0 \pm 1.8$‰ in predators.

Regarding the $^{15}$N enrichment across the different levels of the micro-food web, in grassland, omnivores may act as predators of fungal feeders as indicated by a $\Delta^{15}$N of $3.2 \pm 3.2$‰. In contrast, predators showed a trophic linkage to bacterial feeders with a $\Delta^{15}$N of $3.3 \pm 1.0$‰. In agroforest, however, omnivores were neither related to microbial grazers (bacterial feeders: $0.0 \pm 2.2$‰; fungal feeders: $1.7 \pm 2.8$‰) nor plant feeders ($2.1 \pm 2.6$‰). Comparably, predators showed no link to possible nematode prey.

**Table 2.** Isotopic fractionation of $^{13}C$ [$\Delta^{13}C$ in‰ $\pm$ s.d.] and $^{15}N$ stable isotopes [$\Delta^{15}N$ in‰ $\pm$ s.d] between roots as dominant resource and the different trophic groups of nematodes in the micro-food web of a grassland and agroforest soil.

| Land Use Type | Resource | Plant Feeders | Fungal Feeders | Bacterial Feeders | Omnivores | Predators |
|---|---|---|---|---|---|---|
| | | | | $\Delta^{13}C$ | | |
| Grassland | Plant material root Nematodes | $0.2 \pm 0.7$ | $-0.1 \pm 1.4$ | $-0.3 \pm 1.2$ | $1.2 \pm 0.3$ | $3.5 \pm 0.3$ |
| | Plant feeders | | | | $0.9 \pm 0.5$ | $3.3 \pm 0.9$ |
| | Fungal feeders | | | | $1.3 \pm 1.3$ | $3.7 \pm 1.6$ |
| | Bacterial feeders | | | | $1.4 \pm 1.3$ | $3.8 \pm 0.9$ |
| Agroforest | Plant material root Nematodes | $0.5 \pm 0.5$ | $0.6 \pm 0.4$ | $1.3 \pm 0.3$ | $-0.3 \pm 0.8$ | $1.1 \pm 0.3$ |
| | Plant feeders | | | | $-0.9 \pm 1.0$ | $0.5 \pm 0.4$ |
| | Fungal feeders | | | | $-0.9 \pm 0.5$ | $0.5 \pm 0.3$ |
| | Bacterial feeders | | | | $-1.6 \pm 0.6$ | $-0.2 \pm 0.3$ |
| | | | | $\Delta^{15}N$ | | |
| Grassland | Plant material root Nematodes | $2.1 \pm 2.4$ | $7.8 \pm 4.1$ | $4.8 \pm 1.7$ | $4.0 \pm 1.7$ | $8.1 \pm 2.1$ |
| | Plant feeders | | | | $1.9 \pm 3.4$ | $6.0 \pm 3.8$ |
| | Fungal feeders | | | | $3.2 \pm 3.2$ | $0.3 \pm 4.5$ |
| | Bacterial feeders | | | | $-0.9 \pm 2.5$ | $3.3 \pm 1.0$ |
| Agroforest | Plant material root Nematodes | $4.5 \pm 2.1$ | $4.9 \pm 2.2$ | $6.5 \pm 1.6$ | $6.5 \pm 0.9$ | $9.0 \pm 1.8$ |
| | Plant feeders | | | | $2.1 \pm 2.6$ | $4.4 \pm 1.1$ |
| | Fungal feeders | | | | $1.7 \pm 2.8$ | $4.8 \pm 3.1$ |
| | Bacterial feeders | | | | $0.0 \pm 2.2$ | $3.6 \pm 1.8$ |

### 3.3. Pulse-Labeling with $^{13}CO_2$

### 3.3.1. Incorporation of $^{13}C$ into Soil and Basal Food Web Resources

The incorporation of $^{13}C$ in shoots, roots and the soil after $^{13}CO_2$ pulse labeling varied significantly between these three resources, as well as between these resources and sampling day in grass and willow (Table 3). For both functional plant types, the shoots allocated more $^{13}C$ compared to roots (Table 4), with shoots more enriched in grass than in willow, albeit not significantly. The highest levels of enrichment with $^{13}C$ (expressed in $\log(\Delta^{13}C + 2)$) in shoots were detected at day 3 with $6.6 \pm 0.3$‰ for grass and $5.9 \pm 0.5$‰ for willow, followed by a gradual decrease to a level of $5.6 \pm 0.2$‰ and $5.0 \pm 0.5$‰ until day 28 for grass and willow, respectively. This decrease in $^{13}C$ was significant between day 3 and 28 for willow ($p < 0.05$; Table 4).

**Table 3.** Linear mixed-effect model table of type III error for the effect of basal resources and the soil, sampling day and their interaction on the incorporation of $^{13}C$ into different plant parts and into the soils of grassland and agroforest.

| Land Use Type | Factor | Sum of Square | Mean of Square | df1 | df2 | *F* Value | *p* Value |
|---|---|---|---|---|---|---|---|
| Grassland | res | 175.04 | 87.70 | 2 | 36 | 106.89 | <0.001 |
| | Day | 2.78 | 0.93 | 3 | 36 | 1.13 | 0.35 |
| | res × day | 17.41 | 2.90 | 6 | 36 | 3.54 | 0.01 |
| Agroforest | res | 161.63 | 80.81 | 2 | 29.20 | 214.55 | <0.001 |
| | Day | 0.53 | 0.18 | 3 | 29.26 | 0.47 | 0.71 |
| | res × day | 8.78 | 1.46 | 6 | 29.22 | 3.88 | 0.01 |

df1 numerator degree of freedom, df2 denominator degree of freedom, res resources.

**Table 4.** Enrichment with $^{13}C$ [log($\Delta^{13}C$ + 2) in‰ $\pm$ s.d.] in basal food web resources, i.e., the tissues of grass and willow, and the soil underneath at day 3, 7, 14 and 28 after $^{13}CO_2$ labeling. Values with no or the same letter are not significantly different according to Tukey's HSD test at $p < 0.05$. Letters denote significant differences between $\Delta^{13}C$ between the four sampling days for each basal resource of grass and willow.

| | Day | Grass | Willow |
|---|---|---|---|
| Soil | 3 | $1.1 \pm 0.3$ | $5.6 \pm 0.2$ |
| | 7 | $1.8 \pm 0.9$ | $1.4 \pm 0.7$ |
| | 14 | $1.3 \pm 0.5$ | $1.2 \pm 0.2$ |
| | 28 | $1.2 \pm 0.2$ | $1.3 \pm 0.1$ |
| Root | 3 | $5.0 \pm 0.6$ | $1.0 \pm 1.4$ |
| | 7 | $5.0 \pm 0.6$ | $0.7 \pm 1.3$ |
| | 14 | $4.9 \pm 1.0$ | $2.5 \pm 0.6$ |
| | 28 | $5.7 \pm 0.6$ | $2.1 \pm 1.1$ |
| Shoot | 3 | $6.6 \pm 0.3$ | $5.9 \pm 0.5$ a |
| | 7 | $4.0 \pm 2.5$ | $5.8 \pm 0.1$ ab |
| | 14 | $6.4 \pm 0.3$ | $5.3 \pm 0.3$ ab |
| | 28 | $5.6 \pm 0.2$ | $5.0 \pm 0.5$ b |

In comparison to shoots, the $^{13}C$ incorporation (expressed in log($\Delta^{13}C$ + 2)) into roots was much lower. For grass, the enrichment ranged between $4.9 \pm 1.0$‰ and $5.7 \pm 0.6$‰, while in willow roots, no enrichment with $^{13}C$ was detected. The $^{13}C$ pulse was generally not visible in the soil (Table 4).

3.3.2. Incorporation of $^{13}C$ into Nematode Trophic Groups

The enrichment of nematode tissue with $^{13}C$ varied significantly with trophic group in grassland, showing differences in the incorporation depending on time after labeling (Table 5). The allocation of C derived from recent photo-assimilates was first noticeable at day 3 in plant feeders with $\Delta^{13}C$ (log($\Delta13C$ + 2)) values of $3.0 \pm 1.0$‰, and in omnivores with $3.2 \pm 0.4$‰ (Figure 2). This was significant in plant feeders as well as omnivores compared to the $^{13}C$ signal in fungal feeders ($p < 0.01$), bacterial feeders ($p < 0.05$), and predators ($p < 0.001$). The incorporation of plant-derived $^{13}C$ in plant feeders and omnivores peaked at day 7, followed by a slow decrease until day 28. The allocation of recent photo-assimilates from grass over time was lower in bacterial feeders, omnivores and predators, which was significant at day 14 for fungal feeders and predators compared to plant feeders ($p < 0.001$) and omnivores ($p < 0.05$). Predators showed the lowest incorporation of root-derived $^{13}C$ of all trophic groups.

**Table 5.** Linear mixed-effect model table of type III error for the effect of trophic group, sampling day and their interaction on the incorporation of $^{13}C$ into nematode tissue in grassland and agroforest soils.

| Land Use Type | Factor | Sum of Square | Mean of Square | df1 | df2 | *F* Value | *p* Value |
|---|---|---|---|---|---|---|---|
| Grassland | tg | 69.08 | 17.27 | 4 | 75.10 | 22.51 | <0.001 |
| | Day | 16.39 | 5.46 | 3 | 75.10 | 7.12 | <0.001 |
| | tg × day | 5.04 | 0.42 | 12 | 75.09 | 0.55 | 0.08 |
| Agroforest | tg | 3.01 | 0.75 | 4 | 69.01 | 1.15 | 0.34 |
| | Day | 0.93 | 0.31 | 3 | 69.10 | 0.48 | 0.70 |
| | tg × day | 6.67 | 0.56 | 12 | 69.04 | 0.85 | 0.60 |

df1 numerator degree of freedom, df2 denominator degree of freedom, tg trophic groups.

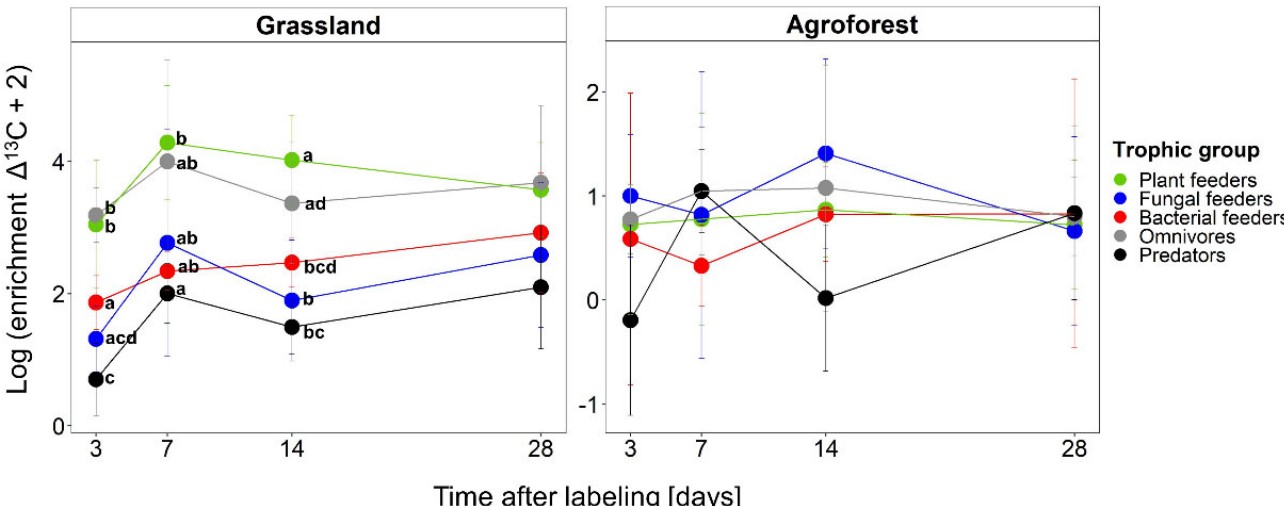

**Figure 2.** Enrichment with $^{13}$C [(log($\Delta^{13}$C + 2) in‰ ± s.d.] of nematode trophic groups at 3, 7, 14 and 28 days after $^{13}$CO$_2$ labeling of grass and willow. Values with no or the same letter are not significantly different according to Tukey's HSD test at $p < 0.05$. Letters denote significant differences between $\Delta^{13}$C between the nematode trophic groups for each sampling day.

In agroforest, the incorporation of $^{13}$C into nematodes did not differ significantly between trophic groups and stayed relatively constant over time (Table 5). As in grassland, all trophic groups (except predators) under willow soil showed a $^{13}$C signal already at day 3, with fungal feeders being most enriched (log($\Delta^{13}$C + 2): 1.0 ± 0.6‰; Figure 2). Predators reached the peak in $^{13}$C allocation at day 7 ($\Delta^{13}$C: 1.0 ± 0.4‰), while fungal feeders and omnivores showed their highest plant C incorporation at day 14 with a $\Delta^{13}$C (log($\Delta^{13}$C + 2)) of 1.4 ± 0.9‰ and 1.1 ± 1.2‰, respectively, yet not significantly.

Trophic groups in grassland incorporated more $^{13}$C than in agroforest (Figure 3), which was significant for plant feeders ($t(35) = 10.9$, $p < 0.001$), fungal feeders ($t(36) = 3.5$, $p < 0.01$), bacterial feeders ($t(37) = 6.8$, $p < 0.001$), omnivores ($t(38) = 9.5$, $p < 0.001$) and predators ($t(36) = 4.3$, $p < 0.001$). The difference in allocated $^{13}$C between plant types was most pronounced for plant feeders and omnivores.

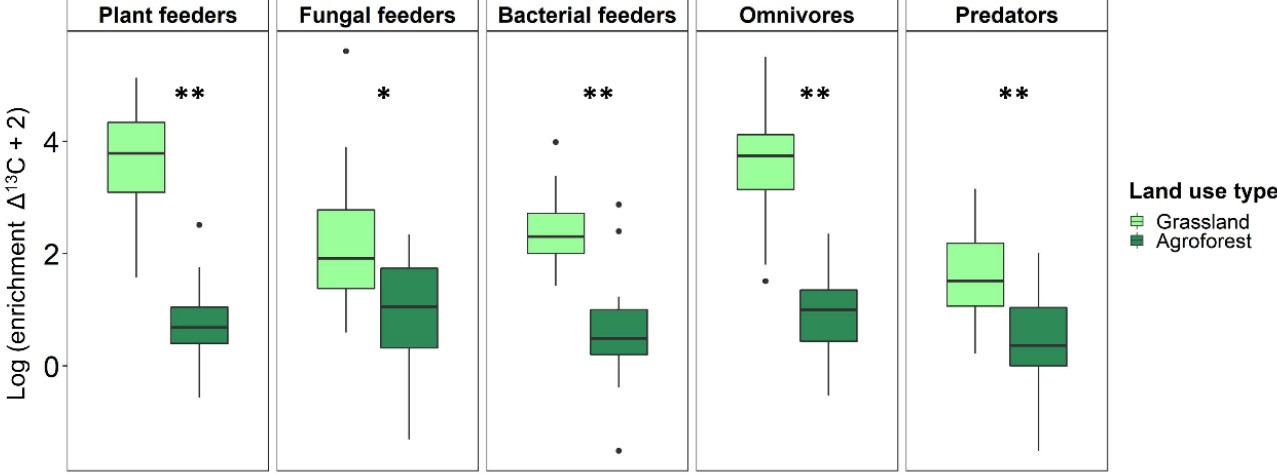

**Figure 3.** Comparison between the enrichment with $^{13}$C [(log($\Delta^{13}$C + 2) in‰] of nematode trophic groups in grassland and agroforest soils. All sampling dates (i.e., 3, 7, 14 and 28 days after $^{13}$CO$_2$ labeling) are bulked together. Two-sample *t*-test with *, ** at $p < 0.01, 0.001$.

## 4. Discussion

### 4.1. Land Use Type Affects The Dominant Soil Carbon and Energy Channel

Grasslands are characterized by a high turnover of shoot and root biomass and consequently by a large pool of organic matter at the soil surface [44]. Further, the grass roots had a low C/N ratio (data not shown). These readily available C resources foster the activity of the bacterial-dominated pathway. Correspondingly, bacterial-feeding nematodes were 18% more abundant under grass than fungal-feeding taxa. In European grasslands bacterial feeders are typically dominant, especially general opportunists of the families Cephalobidae and Plectidae with a high stress tolerance and the ability to survive under food-poor conditions [32,45]. These traits made Cephalobidae also the most abundant nematode family in the investigated grassland soil (Table S1).

The grass–clover mixture of the experimental plots may have additionally enhanced the abundance of bacterial feeders as these are promoted by legumes, while fungal feeders increase under forbs [46]. This is in line with the low *f/b* ratio averaging 0.4, pointing to a bacteria-mediated decomposition in the grassland soil. Similar *f/b* ratios have been frequently reported in the literature (e.g., [47,48]). Viketoft et al. [49] monitored the predominating decomposition pathway as indicated by the Nematode Channel Ratio (NCR, ratio between bacterial and fungal feeders) for eight consecutive years in a grassland established on arable soil. They showed that decomposition was mainly bacteria-dominated, even though fungi became more important after year six, when recalcitrant litter has accumulated. A similar trend was reported by Wasilewska [50], who analyzed nematode communities on meadows differing in age (1 to 12 years old): the succession from newly established to permanent meadows led to a shift from bacterial to fungal mediated decomposition, resulting in an overall increase of the *f/b* ratio with meadow age.

Compared to grassland, forest soil food webs are based on resources with low quality due to the high cellulose and lignin content of leaf litter [51,52]. Accordingly, the C/N ratio of willow twigs and leaves was twice as high as for grass shoots (data not shown). Willow litter contains polyphenols, condensed tannins and glycoside phenols supporting fungal-based decomposition leading to a slow cycling of nutrients [53]. Underlining this, fungal feeders occurred more than twice as often in agroforest than in grassland. This also fits the *f/b* ratio, which was three times higher under willow compared to grass, with a value of 1.3. Similar results were reported by Háněl [54] who observed the activation of the fungal-based decomposition channel after seven years of succession from abandoned arable field to willow shrub and later forest. A high proportion of fungal feeders in the nematode community is typical for forest systems (e.g., [55–57]) and highlights the importance of the fungal-based energy channel under tree cover.

Overall, shifting the arable land to either grassland or agroforest altered the dominance of the soil C and energy channels. The land use change was clearly visible after six years. Culturing willow as short-rotation agroforest led to a switch from bacterial-mediated to fungal-mediated decomposition by accumulation of litter. This changes the conditions in the arable soil from a system with high turnover rates and a rapid nutrient transfer to plants to a slow cycle driven by more complex organic resources [58]. As a result, in agroforest, an increase in soil C sequestration can be expected in the long term.

### 4.2. The Plant Type Affects Predatory Feeding in Higher Trophic Levels

To what extent plant C is incorporated into the belowground trophic network and fosters higher trophic levels is dependent on plant type and material [59,60]. Unlike grass, trees store C mainly in aboveground organs [61]. Mirroring this, willow shoots were considerably more enriched with $^{13}C$ than the roots and the soil. Li et al. [62] suggest that high amounts of unlabeled C in the phloem diluted the signal of $^{13}C$ in roots and more $^{13}C$ is lost via respiration of aboveground plant parts. Depending on their age, tree leaves can act as either a sink or a source of C and may have additionally affected the C flux between tree and soil. On the other hand, the high turnover of willow fine roots should have stimulated the C flux into the soil leading to an enrichment with organic C [53].

Nevertheless, willows have a high demand on nutrients, particularly when cultivated at high densities [53]. The resulting low availability of basal food web resources in willow soils suppresses bottom-up processes, which can affect food web structure and function.

As a $\Delta^{13}C$ of 1‰ between diet and consumer indicates the consumption of this food source, while a $\Delta^{15}N$ of 3.4‰ indicates the trophic position of the consumer [63,64], the following patterns were deciphered: in grassland soil, the natural $\delta^{13}C$ signatures of omnivores and predators differed clearly from each other, indicating that omnivores do not have the same diet as predators. Moreover, the $\delta^{15}N$ was on average 4.1‰ higher than that of omnivores, i.e., predators were one trophic level above omnivores. Thus, the enriched soil conditions, i.e., accumulation of C and nutrients under grass did not foster predatory behavior in omnivores. Rather, the main resources were roots, as suggested by a $\Delta^{13}C$ of 1.2‰ and $\Delta^{15}N$ of 4.0‰ between omnivores and roots. Presumably, the high number of fine roots of grass can be easily accessed and pierced by their dorylaimoid odontostyle. Supporting this, feeding on plants by different omnivore genera has been previously reported, e.g., *Aporcelaimus*, *Dorylaimellus* and *Eudorylaimus* [42,65], with the latter taxon also detected in the grassland soil of the present study (Table S1).

Surprisingly, predators showed no connection to potential nematode prey, i.e., plant feeders and microbial grazers under grass, as $\Delta^{13}C$ exceeded 3‰. Other resources might have been enchytraeids, collembolans, protozoa and rotifers as they are abundant in grassland soils [44,66,67].

In the agroforest soil omnivores had significantly lower $\delta^{13}C$ values than predators indicating different C sources. Further, a low $\Delta^{15}N$ between omnivores and microbial grazers as well as plant feeders points to a non-predatory diet. Contrasting with this, Kudrin et al. [68] found predatory behavior in omnivores in a spruce forest. In this survey, omnivorous Dorylaimida were enriched by ca. 2‰ and 4‰ compared to bacterial and fungal feeders. However, under willow, omnivores were not only separated from other nematode groups as potential prey, but also from roots by a weak depletion in $^{13}C$ of $-0.3$‰ and a distinct enrichment of $\delta^{15}N$ by 6.5‰. For Leptonchidae, which occurred in the agroforest soil, fungi were reported as part of the diet [42]. An additional resource might be algae, which are a well-documented food source for omnivorous nematodes [69,70]. Feeding on algae with $^{13}C$ values below $-30$‰ [71] would further explain the distinctly lower $\delta^{13}C$ values of omnivores in two of the five samples investigated for natural isotope ratios (see Figure 1).

Under willow, predators mainly consumed plant- and fungal-feeding nematodes as indicated by a $\Delta^{13}C$ around 0.6‰ and a $\Delta^{15}N$ around 4‰. Bacterial feeders were no frequent food source as they showed an isotopic signal comparable to predators with a $\Delta^{13}C$ of $-0.2$‰ only. This finding is interesting as bacterial and fungal feeders were equally abundant in willow soil. Therefore, bacterial feeders might be an unattractive food source, e.g., because of a lower energy content. Supporting this assumption, it has been reported that the bacterial feeder *Plectus* has less calories than the fungal feeder *Aphelenchus* [72,73] which both were present in the agroforest soil (Table S1).

### 4.3. Plant C Flux Is Strongest and Fastest into Basal Trophic Levels

Enrichment with $^{13}C$ in the nematode tissue was already measurable three days after labeling. Thus, the micro-food webs in both grassland and agroforest were driven by recently fixed plant C. This corresponds with other studies, which detected C enrichment in nematodes two days after labeling with maize as crop [74,75].

As herbivores rely directly on the root as a resource, they are enriched with plant-derived C first [26]. Accordingly, a fast and significant enrichment of $^{13}C$ in plant feeders occurred under grass. In contrast, in the agroforest soil, plant feeders received plant-derived C slower. Likely, the tougher roots of willow trees are more difficult to penetrate for plant-feeding nematodes compared to grass. Moreover, the nematode community under willow comprised only few plant-parasitic nematodes, but rather more members of the Tylenchidae. These taxa are facultative root and fungal feeders [42] and can change

resources if one diet becomes limited [76]. Moreover, willow forms ectomycorrhiza, which can mechanically protect roots against sucking parasites [77]. Supporting this, Háněl [78] suggests that the tylenchid taxon *Filenchus* feeds on mycorrhiza fungi rather than roots in spruce forest soil.

Compared to plant feeders $^{13}$C incorporation into microbial grazers was slower in grassland soil. As bacteria are fast in assimilating root exudates [79,80], bacterial grazers were expected to mirror this by fast enrichment in $^{13}$C. This is especially with regard to the high root biomass of perennial grasslands and thus, great availability of plant C due to exudation and rhizodeposition [81]. Further, the *f/b* ratio indicates that a large portion of C is channeled via the bacterial pathway. However, fungal feeders exhibited an enrichment pattern similar to bacterial feeders. It is likely that, in grassland, fungal feeders are associated with arbuscular mycorrhiza (AM) fungi allocating C directly from their plant host, rather than with saprotrophic fungi relying more on plant litter. Ostle et al. [82] showed that after $^{13}$CO$_2$ labeling in grassland 5–8% of the fixed $^{13}$C was allocated in AM fungi. Feeding on AM fungi by fungivore nematodes has been reported for members of the families Aphelenchidae and Aphelenchoididae [83,84], both families which occurred in the investigated grassland soil. In contrast to grassland, in an agroforest litter resources become more important. Thus, the relevance of saprotrophic fungi in channeling plant C increases considerably. Accordingly, fungal feeders were the first nematodes that incorporated plant-derived C after labeling.

Due to their higher trophic position in the nematode micro-food web, omnivores and predators are supposed to incorporate plant C with a considerable time lag. Correspondingly, in agroforest soil, the $^{13}$C signal in predators was recorded four days later than for the other trophic groups. Contrasting with this, the fast enrichment of $^{13}$C in higher trophic levels in grassland soil points to a more rapid transfer of C through the soil food web. Comparably, Strickland et al. [85] reported that labile C propagated within 3 days through a grassland food web, including higher trophic level predators such as mesostigmatid mites.

In sum, the dependence of the nematode micro-food web on photosynthate C as a resource was indicated by the enrichment with $^{13}$C, which was higher for each trophic group in grassland than in the agroforest soil. The predominant role of the herbivore food chain in grassland is further highlighted by plant feeders, as the trophic group with the highest $^{13}$C allocation. Contrasting with this, the transfer of N to higher trophic levels was mediated by the decomposer food chain with bacteria in a central position (unpubl. data). On the other hand, in the agroforest, C is mainly recruited from soil organic matter (SOM), which fuels the decomposer food web, predominantly fungi and fungal feeders. However, the utilization of SOM by saprotrophic fungi is slower, and thus benefits C sequestration and nutrient retention. Yet, slow decomposition also results in less plant-available N in the agroforest soil, increasing competition for such N forms between plants and soil fauna. The strong links between aboveground vegetation and belowground biota act as drivers for ecosystem processes [86], and, in the case of agroforest as a management practice, can enhance regulating (e.g., C storage) as well as provisioning processes (e.g., biomass production).

## 5. Conclusions

The trophic structure of the nematode micro-food web, as well as the C flux within differed between grassland and agroforest six years after management change. As soil and environmental conditions were generally homogenous across the field site, these results highlight the role of the land use type in determining the composition and function of the soil food web. The herbivore food chain was promoted in grassland, while the detritivore food chain dominated in agroforest. Agroforestry is an ecologically sustainable management practice and known to, e.g., reduce evaporation, enhance water infiltration, and improve soil structure. The present study indicates that slow decomposition processes are fostered in agroforests, which can increase soil C sequestration. Therefore, this manage-

ment method should be promoted, especially in view of its potential for a climate change mitigation strategy.

**Supplementary Materials:** The following supporting information can be downloaded at: http://www.mdpi.com/article/10.3390/agronomy12040976/s1, Table S1: Overview of family structure in grassland and agroforest soil. Figure S1: Overview of study site.

**Author Contributions:** Experimental design, Methodology, Data curation, L.S. and J.P.; Formal Analysis, Software, C.H. and Z.L.; Validation, Visualization, Writing—Original Draft Preparation, C.H. and L.R.; Investigation, C.H., L.S. and Z.L.; Resources, L.S., J.P. and L.R; Writing—Review & Editing, C.H., L.R. and J.P; Supervision, L.R. and J.P.; Project Administration, J.P. All authors have read and agreed to the published version of the manuscript.

**Funding:** This research received no external funding.

**Institutional Review Board Statement:** Not applicable.

**Informed Consent Statement:** Not applicable.

**Data Availability Statement:** Not applicable.

**Acknowledgments:** We gratefully acknowledge the support of Jens Dyckmans and Reinhard Langel (Kompetenzzentrum Stabile Isotope-KOSI, Göttingen) with stable isotope analyses in nematodes. Zhipeng Li was funded by the China Scholarship Council (CSC) (201604910550). The article processing charge was funded by the Deutsche Forschungsgemeinschaft (DFG, German Research Foundation)—491192747 and the Open Access Publication Fund of Humboldt-Universität zu Berlin.

**Conflicts of Interest:** The authors declare no conflict of interest.

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
