# Peer review of "Flux of Root-Derived Carbon into the Nematode Micro-Food Web: A Comparison of Grassland and Agroforest"

_agronomy, doi:10.3390/agronomy12040976_

Round 1

Reviewer 1 Report

Dear Authors,
The specific suggestions has been detailed in the reviewed manuscript. Kindly go through it and incorporate it for readers understanding.
In general,
1. Check the language of the manuscript and check the spellings such as agroforestry instead of agroforest.
2. Recheck your abstract, introduction and conclusion.
3. Concise the statistical analysis, conclusion and add spatial mapping for study site.
4. Add some more information about agroforestry in the introduction and delete unnecessary statements.
5. Arrange the references in the journal format.
6. Some portion of introduction should be part of materials and method mentioned in the reviewed manuscript.
7. Don't use scientific name always. Add it in starting of the manuscript and later with common name.
8.  Brief the hypothesis and add the aims/objectives of this study in the introduction.
9. Add textural class and SI unit.
10. Use some recent references in discussion portion
Other suggestions has been specified in the reviewed manuscript itself

Reviewer 2 Report

Dear authors,

Congratulations for your work, one can notice that you worked a lot in all the data processing. It is a very interesting topic and a relevant one to take decisions for agricultural management.

In the attached document I do some suggestions about the incorporation of some schemes and the relevance of certain aspects. I think they will help to present the study and results clearer.

All the best for the next version.

Kind regards. 
